# Extracts of Apricot (*Prunus armeniaca*) and Peach (*Prunus pérsica*) Kernels as Feed Additives: Nutrient Digestibility, Growth Performance, and Immunological Status of Growing Rabbits

**DOI:** 10.3390/ani13050868

**Published:** 2023-02-27

**Authors:** Mohamed Basyony, Amr S. Morsy, Yosra A. Soltan

**Affiliations:** 1Department of Poultry Nutrition, Animal Production Research Institute, Agriculture Research Center, Dokki, Giza 12126, Egypt; 2Livestock Research Department, Arid Lands Cultivation Research Institute, City of Scientific Research and Technological Applications, Alexandria 21934, Egypt; 3Animal and Fish Production Department, Faculty of Agriculture, Alexandria University, Alexandria 21545, Egypt

**Keywords:** antioxidant status, cecal fermentation, cecal microbial counts, fruit kernels, natural feed additives, phytochemicals, weaned rabbits

## Abstract

**Simple Summary:**

The post-weaning period presents various challenges to the continuity of the rabbits’ production. The use of natural plant secondary compounds is among the promising feed additives to enhance the rabbit’s health status and growth performance. Extracts of apricot and peach kernels contain high biological phytochemicals that possess anti-oxidative and anti-microbial activities components, therefore, they and their mixture can be used as feed additives for weaned rabbits. The treatment of growing rabbits with apricot, peach, and their mixture enhanced growth performance. This enhancement was fit with improvements in nutrient digestibility, blood antioxidant indicators, and immune response of growing rabbits treated with the mixture. These results indicate the effectiveness of the mixture of apricot and peach kernel extract as a feed additive for the growing rabbits.

**Abstract:**

This study assessed the effects of the kernel extracts of apricot (AKE; *Prunus armeniaca*) and peach (PKE; *Prunus pérsica*), and their mixture (Mix) on growth efficiency, feed utilization, cecum activity, and health status, of growing rabbits. Weaned male New Zealand White rabbits at six weeks old [n = 84, 736 ± 24 SE g body weight (BW)] were randomly allotted to four dietary groups. The first group received no feed additives (control), the second and third groups received 0.3 mL/kg BW of AKE and PKE, respectively, and the fourth group received a mixture of AKE and PKE (1:1) at 0.3 mL/kg BW (Mix). Results indicated that 2(3h)-Furanone, 5-Heptyldihydro was found in abundance in both extracts, while 1,1-Dimethyl-2 Phenylethy L Butyrate and 1,3-Dioxolane, and 4-Methyl-2-Phenyl- were the most components detected in AKE and Cyclohexanol and 10-Methylundecan-4-olide were found in abundance in PKE. All the experimental extracts enhanced (*p* < 0.05) the growth performance, cecal fermentation parameters, and cecal *L. acidiophilus* and *L. cellobiosus* count, while PKE and the mixture treatments presented the highest (*p* = 0.001) total weight gain and average weight gain without affecting the feed intake. Rabbits that received the mix treatment had the highest (*p* < 0.05) nutrient digestibility and nitrogen retained, and the lowest (*p* = 0.001) cecal ammonia concentration. All the experimental extracts enhanced (*p* < 0.05) the blood antioxidant indicators (including total antioxidant capacity, catalase, and superoxide dismutase concentrations), and immune response of growing rabbits. In general, fruit kernel extracts are rich sources of bioactive substances that can be used as promising feed additives to promote the growth and health status of weaned rabbits.

## 1. Introduction

Apricot (*Prunus armeniaca*) and Peach (*Prunus persica*) are the most important fruits grown and processed in Egypt. Apricot is primarily grown in Mediterranean nations, while it is also grown in Russia and the United States [1,2]. In addition, peaches are the primary species for a variety of cultivars that are widely grown around the world [3]. Thus, there are sustainable abundance amounts of both apricot and peach kernels worldwide, which are mostly neglected; however, these kernels are known to contain several biologically active chemicals (e.g., tocopherols, phenolic compounds, -carotene, vitamin E, oleic acid, linoleic acid, and different phytosterols) that possess antioxidant activities and antimicrobial against specific species, while enhancing the growth of fiber-degrading microbes [4,5]. The natural feed additives derived from fruit kernels or their extracts are being included in animal nutrition on a larger scale due to their outstanding spectrum of these phytochemicals. Natic et al. [6] recorded that kernel fruit extracts, such as apricot, cherry, nectarine, peach, and plum, are frequently utilized as supplements in the diets of both animals and humans. It has been established that most phytochemicals that possess antibacterial and antioxidant properties can promote the growth performance of growing rabbits [7,8]. The cecal microbial ecosystem and digestive processes of growing rabbits are also altered by plant phytochemicals, which help in improving nutrient absorption while enhancing dietary nutrient utilization [9]. Additionally, plant phytochemicals support the animal’s immune system, and antioxidant state while protecting the feed lipids from oxidative damage [10]. 

These findings provided a hypothesis that kernel extracts of apricot, peach, or their mixture can be used as efficient dietary feed additives for growing rabbits, in the context of high possible incidences of health problems and growth depression which can occur during the post-weaning period and limit the overall animal profitability [9,10].

The objective of the current study was to investigate the effects of AKE, PKE, and their combination on growth performance, nutritional digestibility, cecal microbiota, antioxidant activity, and immunological status of growing rabbits.

## 2. Materials and Methods

### 2.1. Fruit Kernel Extracts, and Characterization

The fruit kernel extracts of apricot (AKE) and peach (PKE) were provided commercially from the El Marwa Factory for natural seed extract, El-Nobaria, Behera, Egypt. The phytochemicals components of each extract were identified as described by Soltan et al. [11] using a Thermo Scientific TRACE-1300 series gas chromatography/ mass spectrometry (GC/Mass; Thermo Fisher Scientific Inc., Waltham, MA, USA). The GC/Mass is equipped with a fused silica DB-5 capillary column (30 m × 0.32 mm, 0.25 μm film thickness) and coupled to a Triple Quadrupole Mass (TSQ 8000 Evo; Thermo Sci.). The Mass spectra were scanned at the range of 40–700 amu, and the scan time was 5 scans/s. The chemical components of the experimental AKE and PKE were detected by the combination of retention index data with mass spectra data using the Mainlib library.

### 2.2. Animal Management and Dietary Treatments

Eighty-four weaned New Zealand White male rabbits (*Oryctolagus cuniculus*) at six weeks of age with an initial BW of 736 ± 23.9 SE g were used in this study. All animals were housed in galvanized wire battery cages with standard dimensions (50 × 45 × 40 cm^3^) located in a well-ventilated building. All cages were fitted with manual feeders and automatic systems of nipple drinkers to provide clean normal fresh water continuously. The animals were kept under similar hygienic, and environmental conditions. The relative humidity and ambient temperature during the experimental period were 60 to 65%, and 22 to 25 °C, respectively. The first group assisted as the control and received 0.3 mL/kg BW of distilled water without any feed additives, this was to control the rabbits’ stress and fear when handled for the treatments [12]. The second and third groups received 0.3 mL/kg BW of AKE and PKE, respectively, and the fourth group received 0.3 mL/kg BW of a mixture of AKE and PKE at a 1:1 ratio (Mix). The experimental extracts were orally administered daily for 56 days (8 weeks). The basal diet was formulated and pelleted to meet the nutrient requirements of growing rabbits [13]. The rations were offered to rabbits *ad libitum* and free access to clean water was provided to the rabbits through stainless steel nipples that were fixed in each cage. The ingredients and chemical composition of the pelleted rations are presented in Table 1. The total protein (CP), crude fiber (CF), and ether extract (EE) recorded values of 17.2, 13.1, and 3.45%, respectively. 

### 2.3. Growth Performance and Dry Matter Intake

During the growth trial, all rabbits were weighed individually once a week before the morning feeding to determine the average daily gain (ADG). Feed refusals were collected and weighed daily to determine the dry matter intake (DMI). The feed conversion ratio (FCR) was calculated by dividing the DMI by the ADG.

### 2.4. Nutrient Digestibility and Nitrogen Balance

At 87 days of animal age, we started the complete daily collection of feed refusals, urine, and feces for 5 consecutive days as a collection period. The excreted feces and urine of each cage were collected daily in bags before offering the morning meal. Representative samples of the daily amount of feces were stored at −20 °C and later pooled for each cage. At the end of the digestibility trial, the fecal samples were dried at 70 °C for 48 h in a forced air oven, ground to pass through a 1 mm screen on a Wiley mill grinder, and stored at −20 °C for further chemical analyses. Samples of the offered and residual diets and feces were analyzed chemically for dry matter (DM), organic matter (OM), CP = 6.25 × N, EE, and CF according to AOAC [14]. For the nitrogen balance determination, urine samples were acidified with 10 mL sulfuric acid (H_2_SO_4_; 1M), frozen at −20 °C, and later pooled for each cage. By the end of the collection period, urine samples were thawed, centrifuged (15,000× *g* for 20 min) at room temperature, and analyzed for total nitrogen using the micro-Kjeldahl method [14]. The contents of the non-fiber carbohydrate (NFC) were calculated as NFC (%) = 100 − (CP + Ash + EE + CF).

### 2.5. Blood Plasma Antioxidant and Immunological Indicators

Ten animals were selected randomly from each treatment at age of 10 weeks and injected intramuscularly with 0.5 mL of a suspension of the sheep red blood cell counts (SRBCs; 10% concentration), as a T-dependent antigen [15]. Plasma levels of hemagglutination antibodies against SRBCs were measured at 11, 12, and 13 weeks of animal age using a hemagglutination test, and the plasma antibody titers were calculated as log2 values [16].

The same ten animals were subjected to blood plasma collections (after 12 h of fasting) for measuring the antioxidant biomarkers. At day 92 of the animal age (the last experimental day), the blood samples were harvested from the rabbit marginal ear vein using heparinized tubes (BD Vacutainer^®^ Tubes, Jersey, NJ, USA). The blood samples were centrifuged at 2000× *g* for 20 min at 5 °C to obtain plasma. The collected plasma was stored at −20 °C for further antioxidant analyses. Malondialdehyde (MDA), catalase activity, superoxide dismutase activity, and total antioxidant capacity (TAC) were measured colorimetrically using commercial kits produced by Bio Diagnostic Inc., Dokki- Giza, Egypt.

### 2.6. Cecal Measurements, Fermentation Characteristics, and Microbial Count

On the last day of the experiment (day 92 of age), the ten animals (that were subjected to blood sampling) were slaughtered as described by Lopez et al. [17]. The cecum of each rabbit was removed and weighted (fully and empty). The cecal length was measured with a ruler. The pH of the cecal contents was measured immediately using a digital pH meter (Model 20, Digital pH meter for Orion Research, Alaska, Hawaii, Canada). The cecal contents were strained through three layers of cheesecloth and prepared for subsequent ammonia and short-chain fatty acids (SCFAs) determinations. Ammonia concentrations were measured by steam distillation (UDK 139- Semi-Automatic Kjeldahl Distillation Unit, VELP Scientific Inc., Usmate, Italy) [14]. The SCFAs were determined as described by Palmiquist and Conrad [18] using gas chromatography (GC, model 5890, Hewlett Packard, Little Falls, MI, USA) equipped with a capillary column (30 m length × 0.25 mm inner diameter, 1 m phase thickness, Supelco Nukol; Sigma–Aldrich, Mississauga, Canada), and flame ionization detector (FID).

### 2.7. Statistical Analysis

All the obtained data were analyzed statistically using one-way analysis of variance (ANOVA) with SPSS 11.0 statistical software. Differences among means were determined using the Duncan test. Data were analyzed using the following model: Y_ij_ = U + A_i_ + E_ij_ where U is the overall mean, A_i_ is the effect of dietary treatments; and E_ij_ is the random error. For the growth performance and the nutrient digestibility variables, the statistical repetitions were 7 cages per dietary treatment including 3 rabbits each, as the rabbits were housed in groups (7 cages per treatment, 3 rabbits per cage). For the blood plasma and cecal variables and the digestibility trial, the statistical repetitions were the individual rabbit. Statistical significance was accepted at *p* ≤ 0.05.

## 3. Results

### 3.1. Phytochemicals of the Experimental Feed Additives

The phytochemicals compounds in AKE and PKE determined by GC/Mass analysis are presented in Table 2 and Table 3, respectively. The experimental AKE and PKE contained 15 and 26 phytochemicals with different concentrations and functional groups. Both extracts contained 2(3h)-Furanone, 5-Heptyldihydro in abundance; however, the AKE contained almost double the concentration of this component higher than the PKE. 1,1-Dimethyl-2 Phenylethy L Butyrate, 1,3-Dioxolane, and 4-Methyl-2-Phenyl- were the most components detected in AKE. For the PKE extract, Cyclohexanol, 10-Methylundecan-4-olide, Propanoic Acid, Phenylmethyl Ester, and Butanoic acid, 3-methyl-, butyl ester were the highest detected components.

### 3.2. Animal Growth Performance

Effects of AKE, PKE, or their mixture on rabbit growth performance, feed intake, and feed conversion ratio are presented in Table 4. All the experimental treatments increased (*p* < 0.05) the total final BW, total weight gain, and ADG, and enhanced the feed conversion ratio (FCR) compared to the control group; however, both PEK and the Mix groups had similar outstanding experimental growth performance indicators higher (*p* < 0.05) than the AKE and control groups.

### 3.3. Nutrient Digestibility and Nitrogen Balance

Results of the apparent nutrient digestibility and nitrogen balance of rabbits rated with AKE, PKE, and their mixture compared to control rabbits are shown in Table 5. The mixture treatment presented the highest (*p* < 0.05) DM, OM, CP, EE, and NFC digestibilities compared to other treatments. No differences were detected among the experimental groups in fiber digestibility.

### 3.4. Blood Plasma Antioxidant and Immunological Indicators

Results presented in Table 6 showed the effects of the experimental extracts of the fruit kernels on blood plasma antioxidant indicators and SRBCs of growing rabbits. All rabbits treated with the experimental extracts had reductions (*p* < 0.001) in blood plasma concentrations of MDA, and increases in TAC (*p* < 0.001), superoxide dismutase (*p* = 0.017), and catalase (*p* = 0.019) levels compared to rabbits fed the control diet. All rabbits treated with the fruits kernel extracts had similar higher increases (*p* < 0.05) in antibody titers against SRBCs compared to the control at weeks 11, 12, and 13 weeks of age. 

### 3.5. Cecal Measurements, Fermentation Characteristics, and Microbial Count

Results in Table 7 showed that all the experimental fruit kernel extracts positively affected (*p* < 0.05) the cecal length, and empty and free weights compared with the control group. Cecal pH was reduced (*p* = 0.0009) by all the experimental extracts, while the Mix treatment resulted in the lowest (*p* = 0.01) cecal ammonia concentrations among the experimental treatments. Significant increases (*p* < 0.05) in concentrations of cecal total and individual SCFAs were observed for all rabbits treated with the fruit kernel extracts compared to those fed the control group. Cecal total bacteria, coliform, and anaerobic counts were diminished (*p* < 0.05), while *L. acidiophilus* and *L. cellobiosus* were enhanced (*p* < 0.05) with the experimental extracts compared to the control.

## 4. Discussion

The experimental extracts of AKE and PKE were chosen due to their impressive range of biologically active phytochemicals [19,20]. In the current study, most of the phytochemicals identified either in AKE or PKE were found to have various health benefits and beneficial applications such as antioxidant, anti-microbial, and anti-inflammatory activities [5].

The most obvious commonality between the AKE and PKE extracts is the substantial quantity of a fatty ester named 2(3h)-Furanone, 5-Heptyldihydro. It is noticed that the AKE contained almost double the concentration of this component higher than the PKE. Moreover, PKE contained a wider range of phytochemical components than AKE and considerable concentrations of Cyclohexanol, 10-Methylundecan-4-olide, Propanoic Acid, Phenylmethyl Ester, and Butanoic acid, 3-methyl-, butyl ester were identified, while other bioactive components were identified in abundance in AKE different from PKE. These findings indicated that AKE and PKE may have different bioactive activities from each other and, thus, their mixture might have unique biological effects that differ from the individual extract. Soltan et al. [19,21] reported that the binary and tertiary combinations of different plant secondary metabolites from different plant sources directly enhance the bioactivities of these mixtures through their antioxidant and antimicrobial effects.

The similar outstanding indicators of growth performance (final BW, total weight gain, and ADG), and enhanced FCR found for PEK and the Mix groups compared to other groups might be due to providing certain bioactive compounds in these groups that improve nutrient digestion and absorption. The phytochemicals that exist in the fruit kernels may enhance the gut-intestinal microbial ecosystem, thus helping to improve digestive efficiency and nutrient absorption in rabbits [7,22]. Therefore, the enhancements in all nutrient digestibility by all the experimental fruit kernel extracts could be a result of the combination of antibacterial phytochemicals with energy-yielding nutrients (fatty acid esters) found in these extracts. Such combinations of these components may support bacterial symbiosis and the growth of cecal fermentative species, resulting in increased nutrient digestion [9]. A similar relationship between nutrient digestibility and rabbit health was described by Bovera et al. [23]. Despite the restricted number of experimental rabbits (this was the major limitation in the current study), the positive effects of these extracts on nutrient apparent digestibility would be accompanied by enhancements in animal growth performance and health status. 

Some plant phytochemicals, even if existing at very low levels in a mixture, can interact with each other acting as indifferent, antagonistic, additive, or synergistic agents [24]. It seems that the Mix treatment provided the animals with synergetic bioactive compounds that improve nutrient digestion and nutrient absorption since the Mix group had the highest nutrient digestibility values among all the experimental groups. This suggestion can be proved by the superiority of the Mix group in the body N retention compared to other treatments. Moreover, these results indicate efficient protein digestion, absorption in the total alimentary tract, and high utilization of dietary protein for the growth of rabbits that were treated with Mix extract, as all animals received the same diet with similar feed intakes. Decreasing cecal ammonia concentration in rabbits treated with Mix extract may partly confirm the efficiency of protein utilization. Based on the existing study results, it can be suggested that the Mix extract may enhance ammonia utilization; however, the specific mechanisms by which the Mix treatment reduced cecal ammonia are not clear and, thus, more studies are needed to assess these mechanisms. Generally, ammonia constitutes the largest proportion and causes the greatest harm among all kinds of gases in livestock industries [25]. In this context, the extracts of fruit kernels may provide an advantage for further developing the ammonia emissions reduction approach from rabbits.

Recently, there is growing attention on using alternative natural antioxidants from plant extracts in livestock production [11]. In the current study, the treatments of all fruit extracts resulted in notable enhancements of the blood antioxidant indicators. Fruit kernels have been already employed as a source of plant antioxidants, including apricot, peach, sweet cherry, nectarine, and plum oils [3]. The oral administration of AKE and PKE or Mix resulted in the current findings’ considerable reductions in MDA levels and increases in blood serum antioxidant properties (TAC, superoxide dismutase, and catalase). This may be a result of the phytochemicals naturally found in the kernel extracts. Earlier studies showed a notable decrease in MDA level and an increase in the antioxidant defense level following the administration of the AKE and PKE extracts containing other components than what we detected in the current study. For example, Karadimou et al. [26] reported that PKE contains significant amounts of phenolic compounds and unsaturated fatty acids that are used as active natural antioxidant dietary supplements. Others [27,28] found that the main active components of PKE which play an important role in the regulation of a variety of physical and biological functions were oleic and linoleic acid which aid in preventing the negative effects of free radicals on the body [29]. Kalia et al. [30] related the antioxidant activity of AKE to β-carotene, catechins, neochlorogenic acid, caffeic acid, and flavonoids. These differences might due to the differences in the method of the determination of the phytochemicals rather than the differences in the extraction methods and the plant varieties [11]. Regardless of these differences in phytochemicals responsible for the antioxidant activity of the experimental extracts, the increases in superoxide dismutase can play a key role in protecting body cells from free radicals and oxidative damage [7]. Similarly, the endogenous defense mechanisms, including increases in the TAC, and catalase activities may reduce oxidative damage [31]. Generally, such enhancements of antioxidant status may partly explain the improvements in the growth performance of rabbits treated with the experimental fruit kernels.

The oral administration of AKE, PKE, and Mix extracts increased the antibody titers of growing rabbits against SRBCs. These findings imply that these extracts may enhance the immunological responses of rabbits. There is a wealth of literature reporting a conjugation of enhancements of immunity response with improvements in blood antioxidant status. Adding coconut oil and watercress oil to the diet of growing rabbits improved their immunity synchronized with a decrease in pathogenic cecal bacteria and an improvement in blood antioxidant status [7]. Bendich [32] reported that the presence of carotenoids in AKE can protect phagocytic cells from antioxidative deficiencies, enhance T and B lymphocyte proliferative responses, and alleviate the production of necessary interleukins, resulting in improvement in antibody titer against SRBCs. Similarly, Abdelnour et al. [10] observed that supplementing rabbits with thyme essential oil improved blood antioxidant status while boosting the formation of immunological bodies such as IgM and IgG.

The changes in cecal ammonia concentration and SCFAs profile by the experimental extracts indicated that these extracts can modify the cecal microbial community. The enhancement of total SCFAs as well as acetate, butyrate, and propionate concentrations by the treatments of fruit kernel extracts could partly support their superiority in activating specific populations of cecal microorganisms that are involved in fiber digestion, as SCFAs are the main endproducts of the fiber cecal microbial activity, and enhancement of gut cell development [9,32]. In particular, the increase of butyric acid which known for its positive effect on enterocytes and, in general, on intestinal health [23,33]. Such enhancements of SCFAs may partly explain the improvement in cecal weight and length combined with the pH decrease observed for all kernel extracts treatments compared to the control [33,34]. It is worth noting that decreasing the SCFAs is the typical mode of action of antibiotic growth promoters such as monensin [35]. The active components naturally found in the experimental extracts might also play a role in the characteristics of the cecal fermentation profile since the major phytochemicals identified either in AKE or PKE were found to have antimicrobial activities [5]. The reductions in the total bacterial coliform and anaerobic cecal microbial counts may confirm the antimicrobial activity of these extracts, however, the increases in the total counts of *L. acidiophilus* and *L. cellobiosus* may have been a result of a combined effect of low cecal pH and kernel extracts antimicrobial activity against specific microbes that providing suitable conditions for enhancing the growth of other acidic tolerant microorganisms like *L. acidiophilus* and *L. cellobiosus* [36]. Therefore, it can suggest that these extracts somehow exert similar selectivity against the growth of specific cecal microbial strains while promoting certain other cecal microorganisms. However, these extracts presented nutritional advantages for the growing rabbits, but further. These results indicated the advantages of using these extracts for growing rabbits; however, oral administration may not be the optimal practical method for using such extracts on a commercial scale. This is another limitation of the current study, that has to be considered in our further studies.

## 5. Conclusions

The results of the current study revealed various-faceted biological activities of the phytochemical compounds that naturally exist in the experimental kernel extracts. Oral administration of AKE, PKE, and their mixture for growing rabbits resulted in positive effects on their growth performance, digestibility, nitrogen balance, cecal fermentation, and antioxidant and immune status. The animals treated with the mixture treatment of AKE and PKE had superiority in nutrient digestion and N retention compared to other treatments, thus, synergetic bioactive effects might exist among the compounds of AKE and PKE. Prospective experiments are needed to understand the mode of action of the biologically active chemical compounds in the mixture of these extracts.

## Figures and Tables

**Table 1 animals-13-00868-t001:** Composition (%) and chemical analysis of the basal diet.

Item	Basal Diet
Ingredients (%)	
Berseem clover hay	28.0
Barley	17.3
Corn yellow	17.9
Wheat bran	12.0
Soybean meal	20.0
Molasses	3.00
Di-Ca-Ph	1.00
Sodium chloride	0.30
Vitamin premix ^1^	0.30
Lysine	0.10
Methionine	0.10
Chemical composition	
Organic matter(%)	89.7
Crude protein (%)	17.2
Crude fibre (%)	13.1
Neutral detergent fibre (%)	37.5
Acid detergent fiber (%)	21.4
Ether extract (%)	3.45
Non-fiber carbohydrate (%)	55.9
Hemicellulose (%)	16.2
Calcium (%) *	0.83
Available phosphorus (%) *	0.31

^1^ Contains (unit/ kg diet): Vitamin A, 12,000 IU; vitamin D3, 2000 IU; vitamin E, 11 IU; vitamin K, 2 mg; d-Ca pantothenate, 10 mg; folic acid, 1 mg; choline chloride, 250 mg; manganous oxide, 60 mg; ferrous sulfate, 30 mg; zinc oxide, 50; copper sulfate, 10 mg; ethylenediamine dihydrocodeine, 1 mg; cobalt sulfate heptahydrate, 0.1 mg sodium selenite.* Calculated.

**Table 2 animals-13-00868-t002:** Chemical constituents identified by gas chromatography/mass spectrometry analysis in apricot extract (AKE).

Peaks	Compounds	RT (min)	Peak Area (%)	Molecular Formula	Structure
1	D-Limonene	5.39	1.94	C_10_H_16_	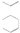
2	Propanoic acid, 2 methyl-, 3-methylbutyl ester	5.93	4.44	C_9_H_18_O_2_	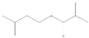
3	1,6-Octadien-3-Ol, 3,7-Dimethyl-	7.00	3.74	C_10_H_18_O	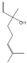
4	Butanoic acid, 3-methyl-, 3-methylbutyl ester	7.09	7.11	C_10_H_2_0O_2_	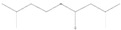
5	Acetic acid, phenylmethyl ester	8.53	6.41	C_9_H_10_O_2_	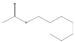
6	1,6-Octadien-3-Ol,3,7-Dimethyl-, Acetate	10.72	1.07	C_12_H_20_O_2_	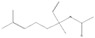
7	1,3-Dioxolane,4-Methyl-2-Phenyl	11.20	10.76	C_10_H_12_O_2_	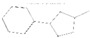
8	Butanoic Acid,Phenylmethyl Ester	13.21	2.21	C_11_H_14_O_2_	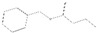
9	Butanoic acid, 2-methyl-, phenylmethyl ester	14.21	1.94	C_12_H_16_O_2_	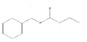
10	2-Buten-1-one,1-(2,6,6-trimethyl-1-cyclohexen-1-yl)-	14.76	0.91	C_13_H_20_O	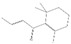
11	2(3h)-Furanone, 5-Hexyldihydro-	16.21	10.32	C_10_H_18_O_2_	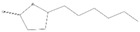
12	3-Buten-2-one, 4-(2,6,6-trimethyl-1-cyclohexen-1-yl)-	16.50	0.66	C_13_H_20_O	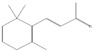
13	1,1-Dimethyl-2 Phenylethy L Butyrate #	16.70	14.36	C_14_H_20_O_2_	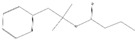
14	2-Phenoxyethyl isobutyrate	17.40	9.10	C_12_H_16_O_3_	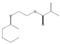
15	2(3h)-Furanone, 5-Heptyldihydro-	18.75	25.01	C_11_H_20_O_2_	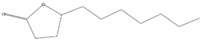

RT = retention time.

**Table 3 animals-13-00868-t003:** Chemical constituents identified by gas chromatography and mass spectrometry analysis in peach kernel extract (PKE).

Peaks	Compounds	RT (min)	Peak Area (%)	Molecular Formula	Structure
1	3-Hexen-1-ol, acetate	4.58	1.45	C_8_H_14_O_2_	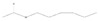
2	Acetic acid, hexyl ester	4.99	1.68	C_8_H_16_O_2_	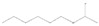
3	Morpholine, 4-(1-butenyl)-	5.12	0.62	C_8_H_15_NO	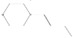
4	1-methyl-4-(1-methylethenyl)-, (S)- D-Limonene	5.39	0.54	C_10_H_16_	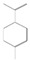
5	Butanoic acid, 3-methyl-, butyl ester	5.72	7.87	C_9_H_18_O_2_	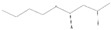
6	Propanoic acid, 2-methyl-,3-methylbutyl ester	5.93	0.28	C_9_H_18_O_2_	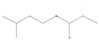
7	1,6-Octadien-3-Ol,3,7-Dimethyl-	7.00	1.38	C_10_H_18_O	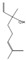
8	Isoamylisovalerate	7.09	1.91	C_10_H_20_O_2_	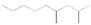
9	Acetic Acid, Phenylmethyl Ester	8.57	1.99	C_9_H_10_O_2_	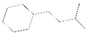
10	Acetic acid, decyl ester	9.74	0.6	C_12_H_24_O_2_	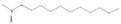
11	Propanoic Acid,Phenylmethyl Ester	10.95	8.64	C_10_H_12_O_2_	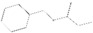
12	1,3-Dioxolane, 4-methyl-2-phenyl-	11.21	7.94	C_10_H_12_O_2_	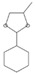
13	Cyclohexanol,5-methyl-2-(1-methylethyl)-, acetate	11.77	14.10	C_12_H_22_O_2_	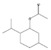
14	Butanoic Acid,Phenylmethyl Ester	13.21	0.83	C_11_H_14_O_2_	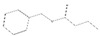
15	Pentanoic acid, phenylmethyl ester	14.21	0.63	C_12_H_16_O_2_	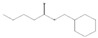
16	2-Buten-1-one, 1-(2,6,6-trimethyl-2-cyclohexen-1-yl)-, (E)-	14.29	0.83	C_13_H_20_O	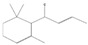
17	2-Cyclopenten-1-One,3-Methyl-2-(2-Pentenyl)-,(Z)-	14.40	2.71	C_11_H_16_O	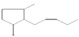
18	(Z)-1-(2,6,6-Trimethyl-1-Cyclo Hexen-1-Yl)-2 Buten-1-One	14.76	1.49	C_13_H_20_O	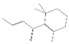
19	2(3H)-Furanone, 5 hexyldihydro-	16.20	7.63	C_10_H_18_O_2_	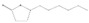
20	1,1-Dimethyl-2 Phenylethy L Butyrate	16.69	4.03	C_14_H_20_O_2_	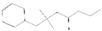
21	2-Phenoxyethyl isobutyrate	17.40	2.40	C_12_H_16_O_3_	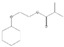
22	2(3H)-Furanone, 5-Heptyldihydro-	18.71	14.99	C_11_H_20_O_2_	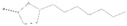
23	10-Methylundecan-4-olide	21.09	9.24	C_12_H_22_O_2_	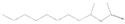
24	ë-Dodecalactone	21.70	1.47	C_12_H_22_O_2_	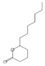
25	9-Octadecenoic acid, methyl ester,(E)-	29.57	0.97	C_19_H_36_O_2_	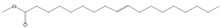
26	9,19-Cyclolanost-24-en-3-ol, (3á)	42.80	3.24	C_30_H_50_O	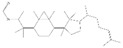

RT = retention time.

**Table 4 animals-13-00868-t004:** Growth performance, feed intake and feed conversion ratio of New Zealand White growing rabbits (36–92 days of age) treated with apricot kernel extract (AKE), peach kernel extract (PKE), or their mixture (1:1).

Items	Treatments				SEM	*p*-Value
		Kernel Extract				
	Control	AKE	PKE	Mixture		
Initial body weight (g/rabbit)	725	741	745	735	26.5	0.977
Final body weight (g/rabbit)	2195 ^c^	2304 ^b^	2437 ^a^	2433 ^a^	25.6	0.001
Total weight gain (g/rabbit)	1470 ^c^	1562 ^b^	1692 ^a^	1697 ^a^	29.8	0.001
Average daily gain (g/day)	26.2 ^c^	27.9 ^b^	30.2 ^a^	30.3 ^a^	0.61	0.001
Feed intake (g/experimental period)	5571	5561	5556	5578	17.9	0.275
Daily feed intake (g/day)	99.4	99.3	99.2	99.6	0.36	0.277
Feed conversion ratio	3.79 ^a^	3.56 ^b^	3.28 ^c^	3.29 ^c^	0.06	0.001

SEM = standard error of the mean. ^a,b,c^ Means with a different superscript in the same row are significantly different (*p <* 0.05).

**Table 5 animals-13-00868-t005:** Nutrient digestibility and nitrogen balance of New Zealand White growing rabbits treated with apricot kernel extract (AKE), peach kernel extract (PKE), or their mixture (1:1).

Item	Treatments				SEM	*p*-Value
		Kernel Extracts				
	Control	AKE	PKE	Mixture		
Nutrient digestibility						
Dry matter	60.0 ^c^	64.3 ^b^	64.4 ^b^	68.8 ^a^	0.622	0.001
Organic matter	62.5 ^c^	64.7 ^bc^	67.6 ^b^	70.8 ^a^	0.647	0.001
Crude protein	69.0 ^b^	70.5 ^b^	70.9 ^b^	73.5 ^a^	0.605	0.008
Ether extract	52.1 ^d^	59.9 ^c^	62.4 ^b^	67.1 ^a^	0.872	0.001
Crude fiber	57.6	58.1	59.2	61.0	1.39	0.293
Nitrogen free extract	65.5 ^b^	66.9 ^b^	69.1 ^ab^	71.8 ^a^	0.97	0.012
Nitrogen (N) balance						
N intake (g/day)	2.77	2.77	2.76	2.77	0.03	0.100
Fecal N excretion (g/day)	1.12	1.093	0.97	0.87	0.02	0.670
Urinary N excretion (g/day)	0.53	0.46	0.48	0.48	0.03	0.315
Body N retention (g/day)	1.12 ^c^	1.21 ^b^	1.31 ^b^	1.42 ^a^	0.04	0.001
N retained (% N intake)	40.4 ^b^	43.8 ^b^	47.5 ^a^	51.3 ^a^	1.03	0.006

SEM = standard error of the mean. ^a,b,c,d^ Means with a different superscript in the same row are significantly different (*p* < 0.05). Intakes of nitrogen, fecal nitrogen, and urinary nitrogen were not affected either by the individual or the mixture of the fruit kernel extracts. The group treated with the mixture presented the highest (*p* = 0.001) body retention of nitrogen while both PKE and the mixture treatments had similar higher (*p* = 0.006) nitrogen retained (as % nitrogen intake) than the other treatments.

**Table 6 animals-13-00868-t006:** Blood plasma antioxidant indicators and antibody titers against sheep red blood cell counts (SRBCs) of New Zealand White growing rabbits treated with apricot kernel extract (AKE), peach kernel extract AKE, or their mixture (1:1).

Item	Treatments				SEM	*p*-Value
		Kernel Extracts				
	Control	AKE	PKE	Mixture		
Antioxidant indicators						
Malondialdehyde (mmol/L)	12.9 ^a^	10.6 ^b^	9.9 ^b^	9.97 ^b^	0.265	<0.001
Total antioxidant capacity (mmol/L)	0.60 ^b^	1.18 ^a^	1.16 ^a^	1.24 ^a^	0.057	<0.001
Superoxide dismutase (U/L)	27.0 ^b^	35.7 ^a^	36.1 ^a^	36.6 ^a^	3.497	0.017
Catalase [U/g]	490 ^b^	595 ^a^	591 ^a^	607 ^a^	28.9	0.019
Antibody titers against SRBCs						
SRBCs at 11 weeks of age	0.73 ^b^	0.89 ^a^	0.88 ^a^	0.86 ^a^	0.025	0.010
SRBCs at 12 weeks of age	0.74 ^b^	0.85 ^a^	0.86 ^a^	0.82 ^a^	0.03	0.044
SRBCs at 13 weeks of age	0.71 ^b^	0.79 ^a^	0.88 ^a^	0.81 ^a^	0.04	0.020

SEM = standard error of the mean. ^a,b^ Means in a row not sharing the same superscript differ significantly (*p* < 0.05).

**Table 7 animals-13-00868-t007:** Cecal measurements, fermentation characteristics, and microbial count of New Zealand White growing rabbits treated with apricot kernel extract (AKE), peach kernel extract (PKE), or their mixture (1:1).

Item	Treatments				SEM	*p*-Value
		Kernel Extracts				
	Control	AKE	PKE	Mixture		
Cecal measurements						
Cecum length (cm)	40.5 ^b^	49.2 ^a^	49.7 ^a^	48.9 ^a^	0.202	0.0001
Full cecum weight (g)	100 ^b^	121 ^a^	120 ^a^	121 ^a^	1.162	0.0007
Empty cecum weight (g)	24.7 ^b^	27.9 ^a^	28.1 ^a^	27.9 ^a^	0.377	0.018
Cecal fermentation						
pH	6.43 ^a^	5.59 ^b^	5.44 ^b^	5.49 ^b^	0.082	0.0009
Ammonia (mmol/L)	13.4 ^a^	11.1 ^b^	11.0 ^b^	9.17 ^c^	0.212	0.001
Total short-chain fatty acids (mmol/L)	51.6 ^b^	69.3 ^a^	69.0 ^a^	70.1 ^a^	3.62	0.05
Butyric acid (mmol/L)	9.1 ^b^	10.3 ^a^	10.2 ^a^	10.5 ^a^	0.165	0.017
Acetic acid (mmol/L)	50.4 ^b^	63.0 ^a^	63.1 ^a^	65.1 ^a^	0.367	0.0001
Propionic acid (mmol/L)	5.12 ^b^	6.27 ^a^	6.23 ^a^	6.4 ^a^	0.09	0.0001
Cecal microbial count (×10^5^) (CFU/mL)						
Total bacterial	4.85 ^a^	3.03 ^b^	3.01 ^b^	2.98 ^b^	0.14	0.0001
Total coliform	3.41 ^a^	2.18 ^b^	2.19 ^b^	2.11 ^b^	0.125	0.005
Total anaerobic	1.91 ^a^	1.22 ^b^	1.21 ^b^	0.91 ^b^	0.27	0.0015
*L. acidiophilus*	1.85 ^b^	8.95 ^a^	8.99 ^a^	9.01 ^a^	0.132	0.0179
*L. cellobiosus*	1.41 ^b^	9.98 ^a^	9.10 ^a^	9.26 ^a^	0.08	0.015

SEM = standard error of the mean. ^a,b,c^ Means within a row having different superscripts are significantly different (*p* < 0.05).

## Data Availability

The data presented in this study are available on request from the corresponding author.

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
