# Peer review of "Extracts of Apricot (Prunus armeniaca) and Peach (Prunus pérsica) Kernels as Feed Additives: Nutrient Digestibility, Growth Performance, and Immunological Status of Growing Rabbits"

_animals, 2023, doi:10.3390/ani13050868_

Round 1
Reviewer 1 Report
Dear Author,
your manuscript shows some element of novelity, considering the use of natural cmpounds in rabbit nutrition to improve the sustainability of this kind of production.
However, several changes must be applied before your manuscipr can be suitable for publication.
The main point is an ethical statement for the animal care along the trial, also considering the way of the administration for the supplemets that you defined as a "gavage" (not a nice word in terms of welfare".
The simple summary must be re-written. This is not a short abstact but it is intended to explain the general idea of your work to a non -scientific staff.
L. 27: please, report the animal age
The introduction must be re-organised: it is too much focused on plant global and local production and low on the effects of the plants in animal nutrition.
L. 100: why "distilled" water?
L. 103. is this procedure in accordance with animal welfare?
L. 107: why "rations"? In the table there is only 1 diet.
L. 109: please, supply methods for chemical-nutritional evaluation of the diet.
Tab. 1: Is digestible energy value calculated? If yes, please report the method. If this is a result, delete from the table and report the value in the results section.
L. 126: correct "trail"
L 144 and L 153 specify "then animals per group"
L 261-262: rewrite this period
L 286 add "A similar relationship between nutrient digestibility and rabbit health was described by Bovera et al 2010" (doi: 10.4995/wrs.2010.18.02)
L 353: add "In particular, the increase of butyric acid is very interesting because its positive effect on enterocytes and, in general, on intestinal health (Bovera et al, 2012; Bovera et al 2010). doi: 10.2527/jas.2011-4119; 10.1017/S1751731110000558
Author Response
Response to the comments of Prof.Dr. Reviewer no 1:
Comment: your manuscript shows some element of novelty, considering the use of natural compounds in rabbit nutrition to improve the sustainability of this kind of production.
However, several changes must be applied before your manuscript can be suitable for publication.
Response: Dear Prof Reviewer, we are thankful for your time and valuable comments that improved our manuscript.
Comment: The main point is an ethical statement for the animal care along the trial, also considering the way of the administration for the supplements that you defined as a "gavage" (not a nice word in terms of welfare".
Response: Please accept our apologies for using this term, we modified this sentence in the revised manuscript, and we replaced it with orally administrated. Please note that oral administration is a common practice to be sure that the treatment was successfully received by the animal, also many works of literature are using oral administration for rabbits for the same purpose, here is an example of the recent articles published by Animals using the oral administration for rabbits.
Animals 2022, 12(11), 1401; https://doi.org/10.3390/ani12111401
Also according to the regulations of the journal, all proposals have to be accepted by the institution review board and we confirm that the animal study protocol was approved by the Institutional Review Board (or Ethics Committee) of the Pharmaceutical and Fermentation Industries Development Center of the City of Scientific Research and Technological Applications (SRTA-City) - (protocol code IACUC76-1A-0123 ).
Comment: The simple summary must be re-written. This is not a short abstact but it is intended to explain the general idea of your work to a non -scientific staff.
Response: Done as advised.
Comment:L. 27: please, report the animal age
Response: we already reported the animal age, so no modifications were done here.
Comment: The introduction must be re-organised: it is too much focused on plant global and local production and low on the effects of the plants in animal nutrition.
Response: Done as advised.
Comment: L. 100: why "distilled" water?
Response: to control the stress effects of rabbits, because they are sensitive animals thus most rabbits are fearful when handled, thus to avoid any effects related to the handling we treated the animals with distilled" water, as this type of water is not contain soluble salts as the tab water, that may affect the salt intake if we administrated the animals with tap water. we just tried to keep similar handling conditions for all animals. we added some clarification for this point and we added a suitable reference for that also.
Comment: L. 103. is this procedure in accordance with animal welfare?
Response: yes, all the rabbits were kept under similar management, hygienic and environmental conditions throughout the experimental period.
Comment: L. 107: why "rations"? In the table there is only 1 diet.
Response: we believe that this word is correct here, it means portions, yes we used only one basal diet but this word here means portions of the diet. Rations meaning is different from diets!!
Comment: L. 109: please, supply methods for chemical-nutritional evaluation of the diet.
Response: we already mentioned the full chemical composition in lines 122 to 130.
Comment: Tab. 1: Is digestible energy value calculated? If yes, please report the method. If this is a result, delete from the table and report the value in the results section.
Response: We are thankful for your comment, it was calculated, actually we have just one basal diet, so it dose not make sense to keep these data, especially since we did not use it in the discussion nor the results so we removed this raw.
Comment: L. 126: correct "trail"
Response: Please accept our apologies, it is corrected now.
Comment: L 144 and L 153 specify "then animals per group"
Response: this comment is not clear to me, there is no mention of "then animals per group", but we added ten animals, and we already specified what they are in each mention.
Comment: L 261-262: rewrite this period
Response: I am afraid that I don't understand this comment, there is no period here, I guess you mean this sentence, is it right? so we reperformed the sentence to be more understandable.
Comment: L 286 add "A similar relationship between nutrient digestibility and rabbit health was described by Bovera et al 2010" (doi: 10.4995/wrs.2010.18.02
Response: The authors are grateful for the valuable reference which improved our discussion, done as advised.
Comment: L 353: add "In particular, the increase of butyric acid is very interesting because its positive effect on enterocytes and, in general, on intestinal health (Bovera et al, 2012; Bovera et al 2010). doi: 10.2527/jas.2011-4119; 10.1017/S1751731110000558.
Response: The authors are grateful for the valuable reference which improved our discussion, done as advised.
Again many thanks Prof Dr. Reviewer no 1 for your valuable and accurate comments.

Reviewer 2 Report
The most controversial and dountfull part is 2.2 Animal Managment and Dietetary Treatments
You wrote that there were 3 experimental groups and one control group. Control group were administrated with distilled water........why? why not used water that were in drinking system? distillation removes minerals so this is not "normal water" that should be in a control group - this is another experimental group.......please explain this
First please describe in few words animal housing condition - don't send readers to your other publication - especially that if someone is working from home and may not have thier institution acccess to journals he will only see abstract and sections snippets of your manuscript - Hashen et al., 2019 Livestock Sci
In mentioned above manuscript you invetsigated additive of Moringa Orfilera for rabbits and you added it to drinking water and set experimantal concentration of this additives in drinking water!!!! why here you were adding it via oral administration?? - noone will do it this way especially in commercial rabbit breeding.
In line 123 and 124 (part 2.4 Nutrient Digestibility ....) you wrote "The excreted feces and urine of each cage were collected daily in bags before offering the morning meal" and in 2.2 in line 105 there is ad libitum feeding. please specify this l"
Table 1 - please add separation - e.g. if you sum everything under Chemical composition till Digetsive enegry you will get over 140% which is impossible and missleading readers. Also please add in text toal protein content%, cerude fibre % and fat % of feed (it should be on the package of feed)
best regards
Author Response
Response to the comments of Prof.Dr. Reviewer no 2:
Comment: The most controversial and dountfull part is 2.2 Animal Managment and Dietetary Treatments
Response: Dear Prof Reviewer, we are thankful for your time and valuable comments that improved our manuscript. Concerning this part, we followed all of your suggestions and modified the text as well.
Comment: You wrote that there were 3 experimental groups and one control group. Control group were administrated with distilled water........why? why not used water that were in drinking system? distillation removes minerals so this is not "normal water" that should be in a control group - this is another experimental group.......please explain this
Response: This sentence is modified now for more understanding. All the animals were drinking normal water including the control group (Ad libitum) as we mentioned in our first version of this manuscript, the distilled water was only treated for the control group, as they received the same quantity of the extract but distilled water just to control the stress that was caused by the handling of the animals while they were orally administrated with the extracts because rabbits are sensitive animals thus most rabbits are fearful when handled, thus to avoid any effects related to the handling we treated the animals with distilled" water, as this type of water does not contain soluble salts as the tab water, that may affect the salt intake if we administrated the animals with tap water. we just tried to keep similar handling conditions for all animals. we added some clarification for this point and we added a suitable reference for that also.
Also, many works of literature are using oral administration for rabbits for the same purpose, here is an example of the recent articles published by Animals using oral administration for rabbits.
Animals 2022, 12(11), 1401; https://doi.org/10.3390/ani12111401
Comment: First please describe in few words animal housing condition - don't send readers to your other publication - especially that if someone is working from home and may not have thier institution acccess to journals he will only see abstract and sections snippets of your manuscript - Hashen et al., 2019 Livestock Sci
Response: We are thankful for your comment, yes done as advised.
Comment: In mentioned above manuscript you invetsigated additive of Moringa Orfilera for rabbits and you added it to drinking water and set experimantal concentration of this additives in drinking water!!!! why here you were adding it via oral administration?? - noone will do it this way especially in commercial rabbit breeding.
Response: Thanks for the comment, we orally administrated the animals with the extracts to be sure that the animals received the experimental doses. Yes, we are agreeing with you that for commercial breeding may oral administration is not a practical way, thus we mentioned this issue as one of the limitations of this study.
Comment: In line 123 and 124 (part 2.4 Nutrient Digestibility ....) you wrote "The excreted feces and urine of each cage were collected daily in bags before offering the morning meal" and in 2.2 in line 105 there is ad libitum feeding. please specify this l"
Response: yes the feeding was ad libitum and was not restricted, now we mentioned the housing conditions as you suggested in your first comment, that the cages were fitted with manual feeders and automatic systems of nipple drinkers to provide clean normal fresh water continuously, the animals were received a known quantity of the diets manually hat allowed to have always refusals, and during the digestibility, we have calculated the collected refusals of the diet.
Comment: Table 1 - please add separation - e.g. if you sum everything under Chemical composition till Digetsive enegry you will get over 140% which is impossible and missleading readers. Also please add in text toal protein content%, cerude fibre % and fat % of feed (it should be on the package of feed)
Response: Data in the original version of this manuscript in table 1 is already separated into two sections that can not be summated, (Ingredients and chemical composition), and if you calculate the ingredients you will get 100%, for the chemical composition we can not sum them, e.g, NDF, ADF, and ADL, they are fiber fractions (NDF = cellulose + hemicellulose +lignine), ADF is (cellulose + lignin), so the sum of these data never will be 100, also the NFC (%) = 100 − (CP + Ash + EE + CF), in this case, the sum of (CP + Ash + EE + CF)+ NFC should be 100 as already written in the original version of this manuscript so no modifications done in table 1.
We added the in the text the total protein content%, crude fiber %, and fat % of feed as suggested.
We hope we answered all of your concerns. Again many thanks Prof Dr. Reviewer no 2 for your valuable time and comments.

Round 2
Reviewer 2 Report
Dear authors,
thank you for answears - unfortunately I still have seriuos doubts about distilled water administration for control group. As I was teached if I want to have a contorl group - it is group that is kept under most standard condition without any different additives/housing conditions etc. And have "normal" feed - so no additives or some feeding restrictions and also have unlimited acces to water similar to other groups
Author Response
Response to the comments of Prof.Dr. Reviewer no 2:
Dear Prof Dr. reviewer, this water has nothing, we do the same as you believed, we have to control all the conditions among the experimental groups, the extracts were administrated orally to the animals, so they may get stressed and scared from the handling, and this stress may interact with the obtained results, so we have to treat the control with the distilled water by the exact handling with the same person to adjust all experimental conditions among the experimental groups. If we did not treat the control group with distilled water, may we will get effects related to the stress that was caused by the handling of the animals, as the treatments groups were orally traded with the treatments (we mentioned in our first revised manuscript for this limitation as adding these extracts may be better in the water directly). Actually, the control received no further additives, (this water has nothing) and all animals have the same drinking conditions of fresh water, so no additives or some feeding restrictions and also have unlimited access to water similar to other groups as you believed. Another important issue is that the use of distilled water in science projects assures that the outcome of the test is fair. Because distilled water basically contains nothing in it, since it is inert, it won't affect the outcome of tests completed for science projects. As a control element, when conducting multiple science projects or tests, pure water won't change the results of the test. If there were minerals or live organisms in the tab water, this could lead to results that are not fair, but biased, which means the results are not accurate if we use the tab water (https://sciencing.com/distilled-good-control-science-projects-7418493.html). The distillation process also removes the electrical charge from the atoms and molecules in water. As we replied to your concern before, the distilled water was only treated for the control group, as they received the same quantity of the extract but distilled water just to control the stress that was caused by the handling of the animals while they were orally administrated with the extracts because rabbits are sensitive animals thus most rabbits are fearful when handled, thus to avoid any effects related to the handling we treated the animals with distilled" water.
Also, many works of literature are using oral administration for rabbits for the same purpose in the control group, here is an example of the recent articles published by Animals using oral administration for rabbits.
Animals 2022, 12(11), 1401; https://doi.org/10.3390/ani12111401
not only The literature in many journals reported the administration of distilled water as a common practice to the control groups for other sensitive animals (like milking sheep and goats or in late pregnancy)
please see https://doi.org/10.1080/10495398.2021.1898978
https://doi.org/10.1016/j.ijpddr.2020.11.004
Animals 2021, 11(10), 2797; https://doi.org/10.3390/ani11102797
We modified sentence line 99 to be more clear for the reader.
We hope we answered all of your concerns. Again many thanks Prof Dr. Reviewer no 2 for your valuable time and comments.
